# Public Health Impact of Using Biosimilars, Is Automated Follow up Relevant?

**DOI:** 10.3390/ijerph18010186

**Published:** 2020-12-29

**Authors:** Antoine Perpoil, Gael Grimandi, Stéphane Birklé, Jean-François Simonet, Anne Chiffoleau, François Bocquet

**Affiliations:** 1Compliance Department, Amgen SAS, 92100 Boulogne-Billancourt, France; antoine.prp@outlook.fr (A.P.); jsimonet@amgen.com (J.-F.S.); 2Faculty of Pharmaceutical and Biological Sciences, University of Nantes, 44035 Nantes, France; gael.grimandi@univ-nantes.fr (G.G.); Stephane.Birkle@univ-nantes.fr (S.B.); 3University of Nantes, INSERM UMR1229, RMeS, Regenerative Medicine and Skeleton, ONIRIS, 44322 Nantes, France; 4Central Pharmacy, University Public Hospitals of Nantes, 44093 Nantes, France; 5Université de Nantes, CRCINA, F-44000 Nantes, France; 6Sponsor Department, University Public Hospitals of Nantes, 44093 Nantes, France; anne.chiffoleau@chu-nantes.fr; 7Law and Social Change Laboratory, Faculty of Law and Political Sciences, University of Nantes, CNRS UMR6297, 44300 Nantes, France; 8Oncology Data Factory and Analytics Department, Institut de Cancérologie de l’Ouest, 44800 Nantes-Angers, France

**Keywords:** machine learning, biosimilars, immunogenicity, economic incentives, safety

## Abstract

Biologic reference drugs and their copies, biosimilars, have a complex structure. Biosimilars need to demonstrate their biosimilarity during development but unpredictable variations can remain, such as micro-heterogeneity. The healthcare community may raise questions regarding the clinical outcomes induced by this micro-heterogeneity. Indeed, unwanted immune reactions may be induced for numerous reasons, including product variations. However, it is challenging to assess these unwanted immune reactions because of the multiplicity of causes and potential delays before any reaction. Moreover, safety assessments as part of preclinical studies and clinical trials may be of limited value with respect to immunogenicity assessments because they are performed on a standardised population during a limited period. Real-life data could therefore supplement the assessments of clinical trials by including data on the real-life use of biosimilars, such as switches. Furthermore, real-life data also include any economic incentives to prescribe or use biosimilars. This article raises the question of relevance of automating real life data processing regarding Biosimilars. The objective is to initiate a discussion about different approaches involving Machine Learning. So, the discussion is established regarding implementation of Neural Network model to ensure safety of biosimilars subject to economic incentives. Nevertheless, the application of Machine Learning in the healthcare field raises ethical, legal and technical issues that require further discussion.

## 1. Introduction

Biologic drugs are complex molecules, produced using a variety of technologies, such as recombinant DNA [1]. These biologics are defined by their production processes and controls. Thus, the regulatory requirements for biologics are more restrictive than those applied to chemical drugs [2]. These requirements establish a regulatory framework for the expensive developments that give rise to biologic molecules [3]. Only a reference biologic drug is initially approved and financial compensation is assured for its development costs. However, after the patent has expired, this commercial protection wears away, allowing other pharmaceutical companies to copy the reference biologic drug. These copies, called biosimilars, are required to be markedly similar to the biologic being copied, without any significant clinical differences [4]. However, although chemically close, biosimilars are not identical to the reference biologic drug [3] contrary to generics. This review proposes to emphasize the complexity of both biosimilars and Machine Learning but also the great opportunity of combining both to generate insights on large population.

## 2. Method

An analysis of the literature was conducted on PubMed and Elsevier for articles published between January 2017 and March 2020. The analysis was focused on Machine Learning applications in healthcare and Biosimilars Safety. The choice of the analysis period was intended to look only at recent articles in this field of Machine Learning where innovation leads to rapid changes. Information gathered were crossed with the latest results from Biologics registries and safety overviews to establish the discussion. 

Seventy-three articles were identified using the following key words: “Healthcare AND AI AND/OR Machine learning”, “Healthcare AND Neural network model”, “Healthcare Data AND Patterns AND/OR correlation”, “Healthcare Data AND classification AND/OR prediction”, “Healthcare Big Data processing”, “Pharmacovigilance AND/OR long-term safety of Biosimilars”, “Biosimilars AND Bio-Originator Safety”, “Biologics Adverse event AND/OR Immunogenicity”, “Shared medical decision AND Interchangeability AND/OR Switch”, “Biosimilar Interchangeability AND Nocebo effect”, “Biologics Healthcare costs”, “Biologics AND Clinical trials AND Genetic Variability”. 

Thirty-seven articles were included as being relevant to the conduct of this review (Figure 1). The first set of articles are related to an overview regarding Biosimilars covering economic incentives, unwanted immune reactions, pharmacological and non-pharmacological negative effect resulting in Nocebo effect, and medical practices such as switch. Second set of articles were included to present supporting data available regarding biosimilar safety thanks to Clinical trials and Real-life Data. Third set of articles were included to cover Big Data and Machine Learning applications in healthcare. A specific focus was made on Neural Network, one type of Machine Learning Model, which was emphasized by literature. 

## 3. Results

### 3.1. Immunogenicity of Biosimilars

#### 3.1.1. Structure of Biologic Drugs

The micro heterogeneity of the chemical structures is currently triggering questions in the healthcare community regarding the impact of such minor variations on clinical responses [1]. In fact, obtaining biosimilar drugs that are strictly identical to the reference biomedicines is difficult. Their structure depends on complex processes such as the production of monoclonal antibodies (mAb) involving immortalised lymphocytes [5]. The very nature of the monoclonal antibody depends on the process implemented. In fact, there is a biosimilar for each production process [2], meaning a particular step might impact the structure of the biologic drug. Unpredictable variations can occur during post translational steps and result in changes to the glycosylation of a protein [1]. These variations, referred to as micro heterogeneity, are observed for each batch of biologic drugs [6], both references and biosimilars. Micro heterogeneity is acceptable for as long as it complies with the regulatory threshold established by the EMA [1]. However, any alteration to the product resulting in significant clinical differences will no longer be accepted [1].

#### 3.1.2. Immunogenicity

Clinical responses may be linked to both wanted and unwanted immune reactions [7]. Unwanted reactions are associated with a negative chain of biological responses that cause damage to the body. They can take the form of fulminant immune reactions or the production of anti-drug antibodies, depending on individual variations [5]. Anaphylaxis and hypersensitivity are two major safety concerns [5]. Immune reactions may also lead to treatment failure.

The immunogenicity of any biologic drug is assessed during its development and real-life use. However, it is difficult to assess. In fact, a single injection may induce antibody responses, referred to as a vaccine reaction [5], while other immune reactions may only be induced after multiple injections over several months of treatment for chronic disease [5]. This delayed clinical response makes it difficult to assess immunogenicity [1]. Thus, it is currently not possible to conclude that monoclonal antibodies administered in multiple injections do not induce immune reactions [5]. 

Moreover, numerous factors may lead to or increase the risk of immune reactions. In fact, the other drugs administered to the patients and the pathologies or populations being treated may influence a patient’s immune response [5]. On the one hand, administering multiple batches of the same biologic drug might increase the risk of immune reactions [1]. Although humanised mAb are less immunogenic, post-translational modifications may also be related to immunogenicity [8]. On the other hand, any concomitant medications prescribed may be involved in inducing an immune reaction [5]. Moreover, medical practice may play a part in this unwanted immunogenicity of biologics [9] regarding “switch” practices.

#### 3.1.3. Nocebo Effect

Alongside unwanted immune reactions, the nocebo effect related to the use of biosimilars should also be mentioned. This nocebo effect is the negative effect of a pharmacological or non-pharmacological treatment [10]. In fact, this effect may result in a poorer quality of life or affect treatment compliance [10]. Certain risk factors have been discussed in the literature, such as the administered dose, the verbal suggestion of arousal or the symptoms or type of a clinical condition [10]. A nocebo effect may occur when switching from a reference drug to its biosimilar, resulting in a loss of efficacy or adverse events [11]. Like immunogenicity, it is possible to assess this nocebo effect, and indeed it should be taken into account when studying any unwanted immunogenicity induced by the use of a biosimilar.

### 3.2. Assessment of Immunogenicity 

#### 3.2.1. Assessment 

In view of the complexity of immunogenicity related to biologics, its assessment is essential during their development. All approved drugs must demonstrate their safety, quality and efficacy in order to obtain approval from the authorities [1]. That is why the potency and immunological profile of biosimilars are assessed to demonstrate their biosimilarity to the reference drug [1,12]. Several assessment steps that have been determined through scientific consensus are implemented as part of the regulatory requirements [4]. First, comparability exercises are carried out, such as the in vitro sequencing of proteins [1]. Further, anti-drug antibody (ADA) assays are performed at an early stage in order to determine the induction of the CD4+T responses that lead to ADA [7]. Pharmacokinetics and pharmacodynamics are then studied in the context of preclinical trials. These are conducted to assess the toxicological profile of the biologic drug by means of cross reactivity studies [13]. Finally, clinical trials are performed to determine the causality between use of the biosimilar and a biological effect under ideal conditions [14]. These clinical trials can demonstrate that there are no significant clinical differences between the biosimilar and reference biologic drugs. 

#### 3.2.2. Limitations of Clinical Trials

However, clinical trials can only offer a limited assessment of the safety of biosimilars in real-life. The patients being may have more comorbidities than those recruited for clinical trials [15]. Moreover, clinical trials focus on populations with little genetic variability, while patients treated in real-life may have greater genetic variability [14]. As a result, rare or delayed adverse events may not be observed and assessed during clinical trials [15]. For these reasons, they cannot provide a full picture of the safety profile when using a drug on the long run. As a matter of fact, complementary data could be used to supplement clinical trial findings [4] regarding the assessment of unwanted immunogenicity, and particularly in terms of extrapolating the indication for biosimilars which benefit only from data collected on the reference biologic [13]. Because this extrapolation of indication needs to comply with safety requirements [16], it will be of value to compensate for the lack of data by assessing the real-life use of the biosimilar (Figure 2).

#### 3.2.3. Real-Life Use of Biosimilars

Chemical molecules and biologics can be used in same therapeutic area, such as oncology [1]. This lack of response of the patient to chemical therapies can lead to the prescription of a biologic. Currently, 40% of biologics are used in oncology [2]. Among several parameters, the patient’s prior treatment forms part of the decision-making process for the physician regarding future prescriptions. As a result, a physician may prescribe the biosimilar rather than the reference biologic but it will be considered to be a switch if the patient was already treated with the reference biologic. Such a switch may be necessary when an allergic reaction has been observed or if there are problems in the supply chain [9]. However, switching means replacing a drug with which the clinical response was known, with another where the clinical response is still unknown [6]. That is why decisions to switch must be made for ethical reason on a case by case basis [17] and be supported by a medical decision that is shared with the patient [6].

#### 3.2.4. Immune Risks of a Switch 

A switch to a biosimilar may be associated with tolerance issues [9], whether it has been motivated by an allergic reaction or supply chain disruptions. In that regard, the healthcare community is more concerned about switching to biosimilars than initiating treatment with them [18]. There have been reports in the literature that there are no significant safety differences caused by a switch [16] between a biosimilar and its reference. However, the British Rheumatology Society recommends gathering more data to validate the possibility of these substitution [18], thus supporting the need for real-life data [4].

### 3.3. Economic Incentives

Biologics may be associated with unwanted immune reactions and a nocebo effect occurring during use of biosimilars. However, approved drugs have demonstrated their compliance with the regulatory requirements and are used according to their Summary of Product Characteristics. Biologics are expensive drugs, so they are associated with economic incentives to use their biosimilars. Indeed, biosimilars have been shown to be 20% to 30% less expensive than their reference biologics [2]. Direct incentives could be implemented by the health authorities to encourage their use [19]. For instance, biosimilars with low market penetration could be targeted by such direct incentives [19]. By being more affordable [1], biosimilars improve access to treatment [4] for patients. 

In Europe, the right to substitution can be discussed at a national level as it is dependent on the jurisdiction of each Member State [4]. Substitutions by pharmacists were recently discussed in the French Social Security Funding Law for 2014 [2]. Substitution has been defined as the right of a pharmacist to deliver a different biologic drug than that prescribed by the physician [9]. However, this right was abrogated by the French Social Security Law in 2020 [20]. This decision could have been motivated for safety reasons.

Economic incentives to use biosimilars and questions from the clinical community regarding unwanted immunogenicity can be incompatible. Real-life data could therefore be used with support from Machine Learning to identify the risk factors that increase immune risks or noncompliance. This assessment could support economic incentives and improve the protection of public health. 

### 3.4. Real-Life Data and Big Data 

#### 3.4.1. Real-Life Data

Real-life data are collected after clinical trials during non-interventional studies on approved drugs that are being used under real-life conditions [14]. These data can be collected on a regular basis by healthcare professionals with respect to treatments, patient information or clinical outcomes [15]. Because clinical trials are expensive [21] and have their limitations with respect to assessment, real-life data can supplement their findings [14]. These real-life data offer an opportunity to observe the paradigm shift regarding prescriptions and medical practices [22]. A focus on assessing the costs of clinical practices and also on exploring medical decision-making [14] can also be realized because these studies are not carried during clinical trials. Moreover, these data could be used to identify predictive response parameters among the various heterogeneous responses to treatment [22].

#### 3.4.2. Big Data in Healthcare

Real-life data can be obtained from numerous sources such as hospital information systems, medical-administrative data issued from public or private payers, etc. [23,24]. Real-life data come under the definition of big data as large quantities are generated, with high heterogeneity and flow [25]. They can be processed to obtain answers to clinical questions concerning the assessment of unwanted immunogenicity in real-life use and the importance of economic incentives. For instance, an analysis could focus on determining whether the concomitant use of biosimilars and other medications might cause an increase in unwanted immunogenicity [15]. Numerous databases could be involved to assess such clinical questions by supplying relevant data from medical records or registries [22]. However, data alone cannot provide information without an appropriate processing methodology [22]. Data can be markedly heterogenic, even between different healthcare establishments [26]. That is why it could be challenging to use a classic statistical model to process such data using multiple variables to produce predictive parameters [24]. AI, and more specifically machine learning, is able to process large quantities of unstructured data [24]. Such methods might therefore be suited to answering questions from the healthcare community with respect to biosimilars and their multiple components.

### 3.5. Applications of Machine Learning in Healthcare

#### 3.5.1. Machine Learning Introduction

From a technical point of view, a program uses a set of instructions to produce an outcome using one aspect of human intelligence. Machine learning, which is one type of artificial intelligence method, refers to programs that can adapt the instructions being used to produce the outcome [27]. Here, the human aspect is its ability to learn and adapt the program’s code to experience [28]. A neural network model is one example of machine learning [28], which is often used to detect patterns in data [29,30]. These patterns correspond to identifying important variables or correlations in the data [26] that are associated with the result produced [27]. Such data patterns can then be used for predictive purposes [29]. Machine learning is also used to predict future outcomes in light of the patterns detected [27,30]. So, these methods can be applied in the healthcare sciences, econometrics or epidemiology [25], where there may be millions of data points for each patient [30]. Machine learning could offer a relevant application for pattern detection and prediction. For instance, it could be developed for use in omics, medical imaging, or digital biomarkers [30].

#### 3.5.2. Learning Process

In order to detect patterns or predict outcomes, a machine learning model requires a learning process to become effective. This learning process is used to compile the instructions that will be used by the model to process the data. In fact, this learning is focused on enhancing the performance of the model so as to produce the correct output that reflects the input data available [28]. When using supervised machine learning, the data are available and already labelled as input and output. The learning steps correspond to comparing the output supplied by the model with real and observed outcomes [28]. By doing this, the model is able to adjust and modify the instructions it implemented to enhance its efficiency in producing the correct output in light of the input data it has processed. The learning steps are conducted using three sets of data called training, validation and test [28]. The goal of these learning steps is to develop a machine learning model that only captures general relationships in the data so that it will produce the correct output for new set of data [28].

#### 3.5.3. Neural Network Involvement 

Neural networks are an example of a machine learning model. They are used to process input data in order to produce output data that comply with the instructions developed during the learning phase. This specific model comprises multiple layers; the first layer, hidden layers, and final layers (Figure 3). Neurons from the previous layer process the input data to transmit output data to the neuron in the next layer [28], which in turn will receive multiple inputs from neurons in the previous layer [28]. This data processing continues through the neural network from the first layer to the final layer to produce a final result such as a classification. The hidden layers are responsible for detecting interactions within the data [28]. The number of hidden layers will determine the depth of the network, linked to its ability to detect complex interactions within the data [28]. This ability makes the Neural Network model more suitable to this data processing compared to others model like decision trees. Further discussion not covered by this article should include suitability of different Machine Learning models based on their characteristics and performance.

## 4. Discussion

Machine Learning could be involved in different stages of the life cycle of biologic drugs. For instance, from development to pharmacovigilance, Machine Learning could be used for data mining or to automate current processes. Indeed, the literature mentions that machine learning is used to automate pharmacovigilance processes [31]. This automation could improve both the detection of safety signals and risk management [31]. In addition, machine learning can also be used to identify relationships between biological terms [32] from medical records or social media screening and extract adverse event data [33] as illustrated with the Word2Vec algorithm [33].The latter application is related to the identification of adverse events using syntactical relationships between words [33].

### 4.1. Research on Risk Factors

As well as process automation and social media extraction, the application of a machine learning model to look for patterns in data should be considered. To some extent, assessing a risk by considering the influence of numerous parameters offers a good example of the application of machine learning. The expected patterns could be identified from input data concerning clinical care or the biologics prescribed, and output data with respect to adverse events and clinical outcomes. This type of application could focus on implementing an algorithm to fit to clinical routine in order to facilitate decision making on issues such as a substitution or dose regulation [7]. Indeed, immune reactions potentially involve numerous parameters that can be associated with various types of data resulting in large quantities and a quite high complexity to process the data. Due to data complexity and volume, Machine learning model is more suited to the task than a standard statistical method [28]. As for the assessment of unwanted immunogenicity, a single clinical question needs to be the starting point for a discussion regarding the relevance of machine learning to the biosimilar environment. For instance, a clinical question might focus on determining whether a substitution increases unwanted immune reactions when biosimilars are used in real-life.

### 4.2. First application of Machine Learning in Biosimilars

This section focuses on the use of a neural network to answer clinical questions. For example, take a patient initially treated with a reference biologic drug and then treated with its biosimilar. In theory, a large quantity of data can be extracted from the medical care provided to the patient, including medical records and clinical outcomes. Any relevant data should be included as input data (e.g., drugs prescribed, indication, biosimilar/reference biologic drug, batch number, manufacturer, and glycosylation rate) and output data (e.g., clinical outcomes, adverse events, and non-compliance). Relevant data such as the severity of any immune reactions should also be included [25]. A supervised model could then be used to identify patterns within these input and output data. 

This supervision should focus on establishing relevant categories of outcomes. In that way, the model would learn to process input data to classify patients in predefined categories. As a result, instructions developed by the model may help emphasize relevant parameters playing a significant role in the classification. That is why categories are defined as a function of expected outcomes related to biosimilar use. For example, a suitable classification could cover the absence of immune reactions, weak immune reactions, ADA production, and fulminant immune reactions (Figure 4). Because this classification might be related to the instructions, this can also influence the pattern detection and predictions made by the model.

Hidden layers detect interactions within a group of data [30]. Because the clinical outcomes are already known at this stage, only the instructions are of value as they provide the interactions or correlations identified in the data. However, such data correlations are interesting to make a medical interpretation in order to assess the risk of unwanted immune reactions. This interpretation could then form part of identifying risk factors for the unwanted immunogenicity of biosimilar use under real-life conditions. 

### 4.3. Transposability

The learning phase focuses on developing a neural network model that is capable of processing new sets of data. This so-called generalisability enables an initial analysis that reveals potential interactions within the data. The medical interpretation can lead to identifying the risk factors involved when switching from a reference biologic to a biosimilar. However, evolutions affecting risk factors might be involved over time. In fact, consideration should be given to using the same neural network to perform the same analysis in response to the same medical question over time. For instance, updated data could be processed by the model to generate new interactions for comparison with those identified in the past. Such new analyses might highlight changes to patterns and interactions that require a new medical interpretation to update the assessment of risk factors.

Furthermore, consideration could be given to using the same neural network model to process other relevant data, such as those related to other biologic parameters, newly approved biosimilars or new practices, as being relevant for a safety profile analysis. This transposability of a model might offer an opportunity to address other relevant medical questions regarding the use of biosimilars in real-life. Transposability is crucial to explore any new components that might be involved in the risk of unwanted immune reactions. That application of the neural network model to new questions might nevertheless be challenging from a technical point of view. New input data can be difficult to process using the instructions already developed by the model. The learning ability of machine learning might not be sufficient to modify these instructions and develop a model that is able to process new data. Therefore, the limits to transposability may require to develop a new model that is better suited to processing specific input data.

### 4.4. Approval of a New Biosimilar

First of all, an initial analysis might focus on assessing the risk when switching from a reference to a biosimilar when only one biosimilar has been approved. In fact, only two products have been covered by such an analysis. A new analysis may therefore be relevant when a new biosimilar appears on the market to explore any pattern changes with this newly approved biosimilar. Indeed, a new biosimilar can impact medical practices and result in indirect changes to the safety profiles of both the biosimilar and reference biologic. There are many potential explanations for this; e.g., better penetration of the hospital market by the new biosimilar. A new analysis by a neural network could search for this type of explanation and, for example, determine whether there was a simple transfer of an adverse event from the reference to the biosimilar drug. 

### 4.5. New Parameters

Another advantage of the neural networks thus developed is the introduction of new and specific data into the analysis. For instance, biological parameters, biomarkers or genetic markers could be selected as being relevant to such an analysis. Then it could be investigated whether such biological parameters are predictive of the clinical response of patients to biologics. As an example, the clinical response of haemophilia A depends on the genetic defect affecting factor VIII [5]. This defect results in a lack of immune tolerance, thus causing an unwanted immune reaction [5]. The introduction of new relevant data might make it possible to process relevant data that were not available in previous databases (e.g., medical records). To some extent, it might increase the relevance of the analysis to determining significant risk factors in clinical responses by applying it to assessing both the safety profile and efficacy of biologics. 

### 4.6. New Practices

The opportunity offered by transposability of the model is replication of the analysis performed by the neural network thus developed in order to compare multiple analyses. This could not only answer the initial medical questions posed but also support the safety of biosimilars in real-life in a context of complex healthcare policies and economic incentives. Payers could therefore increase the economic impact of incentives while ensuring the safety of biosimilars. Moreover, this might offer a starting point for discussions on new medical and pharmaceutical practices. In fact, minor differences may be observed following a substitution, relative to the practitioner or healthcare establishment, but more significant differences might be induced by other practices, such as a substitution by a pharmacist. 

Then, a neural network model might offer a solution to address safety concerns before political decisions are taken with respect to the right of substitution by a pharmacist. The analysis could be performed as part of the precautionary principles of health safety [34], and offer a means to estimate the consequences of treatment by a biosimilar rather than the reference biologic. 

As soon as policy decisions have been made, further discussions could focus on applying neural networks to ensuring the safe use of biosimilars in the context of newly approved practices. This analysis would be carried out as part of the assessment with respect to health safety [34]. However, the neural network design might be revised as the drug delivered by a pharmacist might differ from that prescribed by the physician under the approved substitution scenario. Indeed, there is no guarantee that the neural network model developed initially is able to absorb such data changes. Furthermore, the relevant data needed for the analysis might require the involvement of a new database. In practice, data from pharmaceutical records might be crossed with medical records in order to answer medical questions. In this case, a crossover analysis of several databases should also be discussed as this would be challenging from both the legal and technical points of view.

### 4.7. Prediction Hypothesis 

All previous applications of neural networks focus on identifying data patterns, but this type of model can also be used to anticipate events through prediction. In fact, the instructions developed could be utilised to suggest classifications with respect to input data while the clinical outcomes are still unknown. In this way, the neural network model would be able to suggest a clinical outcome in light of the patient’s data, and thus provide support for medical decision-making while using biosimilars or reference biologics under real-life conditions. 

The determination of eligibility could be a way to benefit from the predictive ability of a neural network. Data could be processed to identify whether a patient can be treated by a substitute drug or not using the previously developed instructions and identified risk factors. This neural network would process the data and produce predictions as a function of each signal received [25]. As this is only a prediction, the results supplied by the model would only be a probability and considerable caution would be necessary with respect to medical decision-making [35]. The neural network would only help physicians to identify situations where a specific event might disturb the risk/benefit ratio of using the biologic. On the other hand, the same neural network could be used following a substitution to monitor any disruptive events and notify the physician. Indeed, the model could support medical decision-making when an immune risk or non-compliance risk is increased. Such a model would therefore offer a means of identifying situations that require adjustments to clinical care in order to prevent adverse events or treatment discontinuation. 

### 4.8. Legal Limitations of Machine Learning

There are several limitations to the application of machine learning to the prescription of biosimilars in real-life. Firstly, it is important to consider the legal framework when focusing on the assessment of safety profiles. In fact, the extracted data could then be processed for other purposes. Compliance with GDPR implies that only the initial purpose should be considered. Literature emphasizes that Blockchain is an opportunity to address Data Privacy concerns by enabling a trustworthy and transparent ecosystem [36]. Moreover, any other data processing using machine learning must have to comply with the regulations applicable in each country. For example, public health regulations in France forbid the collection of individual prescribing data from physicians. The neural network model would therefore have to be designed so that these data could not be supplied indirectly by any processing of big data. Such risks can be mitigated by processing aggregated and anonymised data. Moreover, anonymization would have to be demonstrated as the application of machine learning to big data might weaken anonymization; by the re-identification of an individual [37].

### 4.9. Technical Limitations of Machine Learning

The application of neural networks is mostly based on identifying patterns and interactions. This focuses on determining important variables or parameters that exert a crucial effect on the results generated by the model. These patterns can lead to the identification of significant risk factors, depending on their medical interpretation. This interpretation is in turn dependent on the ability of the neural network to reveal meaningful correlations or interactions. However, neural networks may suffer from a lack of transparency, referred to as the black box, which results in challenging a, interpretation and understanding the mechanisms involved in data processing [29]. As a result, the black box might create conflict between processing by the neural network model and human interpretation [30]. To address this issue, the literature mentions attribution mechanisms to facilitate interpretations [24] which focus on identifying signals that play an important role in generating results.

### 4.10. Ethical Limitations of Machine Learning

Finally, any machine learning application needs to comply with the ethical guidelines laid down by national Ethics Committees. Safety profiles could be assessed using the analysis thus described. Healthcare expenditure could also form part of such an analysis if the relevant components are included. However, machine learning must not result in patient profiling in terms of the costs engaged per patient. It is important to demonstrate that neural networks are not designed to facilitate increases in the costs of insurance policies for patients flagged as having more risk factors. From a collective perspective, risk should be supported by all individuals in a society, thus ensuring a viable system that can support collective healthcare expenses. So, in the example of France, machine learning could only be used to support medical decision-making.

## 5. Conclusions

The application of Machine Learning in healthcare creates a lot of opportunities to support biologics development and research. Such technology should be seen as a tool to learn more about biologic drugs after clinical trials and the medical practices during their uses. However, so that Machine Learning can experience its full development in this application, reflections must be undertaken to analyze the economic and medical context of using biosimilars. There are indeed several limitations which have been described previously that need to be addressed at the earliest stages of any application. However, other key elements should also be discussed, such as the involvement of multiple stakeholders from the public and private sectors. Moreover, the application of Machine Learning to other services and products supplied by the pharmaceutical industry and healthcare providers should also be considered.

## Figures and Tables

**Figure 1 ijerph-18-00186-f001:**
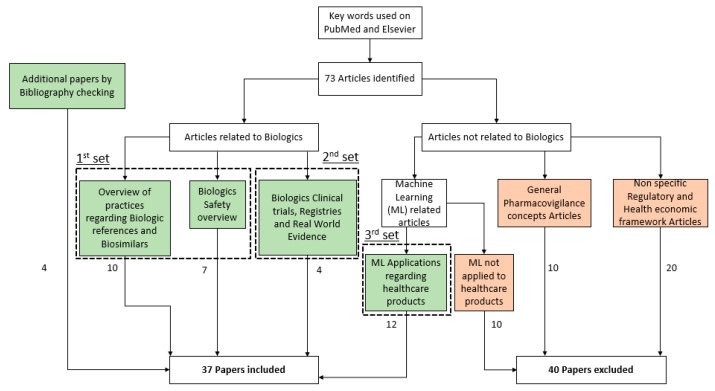
Inclusion and exclusion of identified articles.

**Figure 2 ijerph-18-00186-f002:**
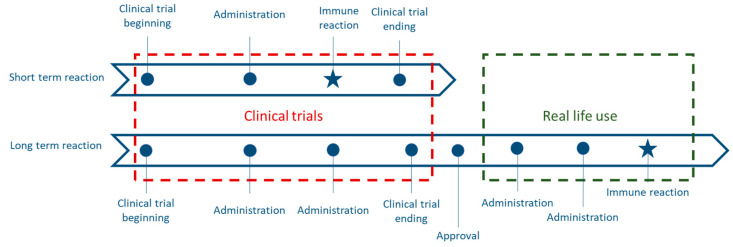
Limitations of clinical trials in terms of assessing long-term reactions.

**Figure 3 ijerph-18-00186-f003:**
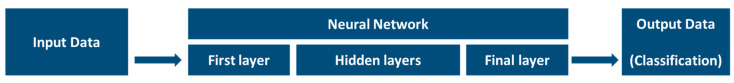
Description of a neural network model.

**Figure 4 ijerph-18-00186-f004:**
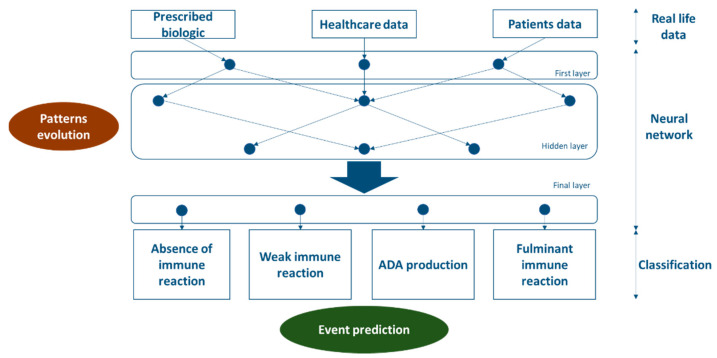
A potentially suitable classification provided by a neural network to assess the immunogenicity of biologic drugs.

## Data Availability

Articles reviewed in this study are available in PubMed and Elsevier.

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
