# Peer review of "Public Health Impact of Using Biosimilars, Is Automated Follow up Relevant?"

_ijerph, 2020, doi:10.3390/ijerph18010186_

Round 1
Reviewer 1 Report
The paper gives an overview of the use of biosimilars, the risks it may entail, and the potential application of artificial intelligence to support safety assessments.
The paper is well structured and well written, the topic is very interesting, but from the point of view of artificial intelligence, it is superficial. As a review of machine learning applied to a problem (biosimilars), the paper should include more references of specific applications of machine learning to that or similar problems, giving the reader a broader view. In fact, of the selected 35 references, many of them are not related to artificial intelligence, so even this figure is smaller. A review paper must give a deeper view of the target, specially in the technical part in this case.
Moreover, the paper focuses on neural networks but there are different techniques of machine learning, some of them which do not have the mentioned limitation of neural networks that is that they are as black boxes, not easily providing explanations about the prediction done.
Why the review has focused on neural networks and no other machine learning techniques?
This should be justified.
I suggest to consider these works:
Jennath, H. S., Anoop, V. S., & Asharaf, S. (2020). Blockchain for Healthcare: Securing Patient Data and Enabling Trusted Artificial Intelligence. International Journal Of Interactive Multimedia And Artificial Intelligence, 6(Special Issue on Artificial Intelligence and Blockchain), 15-23. http://doi.org/10.9781/ijimai.2020.07.002
Redondo, T., Díaz, J., Sandoval, A. M., & Llanos, L. C. (2019). Biomedical Term Extraction: NLP Techniques in Computational Medicine. International Journal Of Interactive Multimedia And Artificial Intelligence, 5(Special Issue on Artificial Intelligence Applications), 51-59. http://doi.org/10.9781/ijimai.2018.04.001
Last, the method section should clearly describe the criteria used to consider an article as relevant.
How many articles were retrieved and subjected to applying the criteria?
Author Response
Dear reviewer, please find our comment in blue :
The paper gives an overview of the use of biosimilars, the risks it may entail, and the potential application of artificial intelligence to support safety assessments.
The paper is well structured and well written, the topic is very interesting, but from the point of view of artificial intelligence, it is superficial. Thank you for your comment. We are aware that this article only addresses partially the involvement of Artificial intelligence regarding Biosimilar use. Our goal was to initiate a discussion about potential applications of AI in order to open further discussion about the relevance of such application. Such applications should be completed by technical reviews of AI applications that we could not cover in this article focused on Biosimilars safety.
As a review of machine learning applied to a problem (biosimilars), the paper should include more references of specific applications of machine learning to that or similar problems, giving the reader a broader view. In fact, of the selected 35 references, many of them are not related to artificial intelligence, so even this figure is smaller. A review paper must give a deeper view of the target, specially in the technical part in this case. Thank you for your comments, We included the 2 articles you mentioned, in the discussion section, as they are nonspecific to Biosimilars in order to provide a broader view and we give more precisions about inclusion and exclusion criteria of identified articles by elaborating the Flowchart 1.
Moreover, the paper focuses on neural networks but there are different techniques of machine learning, some of them which do not have the mentioned limitation of neural networks that is that they are as black boxes, not easily providing explanations about the prediction done.
Why the review has focused on neural networks and no other machine learning techniques?
This should be justified. Thank you for raising this point requiring discussion. As emphasized by Doupe et al, 2019, we decided to mention Neural Network that “Can help predict outcomes with highly complex, nonlinear relationships and interactions”. We decided to mention also that further discussion is required regarding suitability of different AI models.
I suggest to consider these works:
Jennath, H. S., Anoop, V. S., & Asharaf, S. (2020). Blockchain for Healthcare: Securing Patient Data and Enabling Trusted Artificial Intelligence. International Journal Of Interactive Multimedia And Artificial Intelligence, 6(Special Issue on Artificial Intelligence and Blockchain), 15-23. http://doi.org/10.9781/ijimai.2020.07.002 Thank you for mentioning this very interesting article. The information has been included in the discussion part related to data privacy and compliance with GDPR as blockchain is an opportunity.
Redondo, T., Díaz, J., Sandoval, A. M., & Llanos, L. C. (2019). Biomedical Term Extraction: NLP Techniques in Computational Medicine. International Journal Of Interactive Multimedia And Artificial Intelligence, 5(Special Issue on Artificial Intelligence Applications), 51-59. http://doi.org/10.9781/ijimai.2018.04.001 Thank you, we mentioned this article in the discussion part related to information extraction to identify relationship between terms.
Last, the method section should clearly describe the criteria used to consider an article as relevant.
How many articles were retrieved and subjected to applying the criteria? The flowchart has been included to describe inclusion and exclusion of articles resulting in inclusion of 3 sets of articles.
Please do not hesitate to reach out and share your comments.
Reviewer 2 Report
This article discusses the potential applications of AI with respect to immunogenicity or non-compliance. This is an interesting article and topic, but I found it difficult to distinguish between what was background information, what related to the authors analysis of the literature and future directions.
The authors have provided a comprehensive discussion of their topic, but they need to provide:
1) Reasons for their selection of key words in their literature search. For example, "Nocebo effect" is introduced, but only explained later in the manuscript.
2) A flowchart of rationale for their rejection of articles (i.e. they identified 75 articles but their final number was 35).
3) A table of the articles selected, including key information such as authors, title, date, journal, country, ML algorithms used etc
There is a great deal of repetition of information in this manuscript. For example, information about the economic incentives is repeated in the Background and in Section 5.
In Section 5 the authors refer to "Europe" but what about any other continents/countries?
The authors provide a sections on machine learning (7.2) and a section on neural networks (7.3), but section 7.2 is only "an example of a machine learning model". Please provide more examples of the current use with different neural network models (e.g. deep learning (DL) algorithms).
This article seems to be specific to the development of a neural network model and this should be articulated in the title, abstract and aims, rather than the all-encompassing "Machine Learning" term as other algorithms are not investigated.
This is an interesting topic and the proposed model is interesting, but the manuscript would greatly benefit from harmonisation of the information, re-structure of sections to make it clearer to the reader the background (Introduction), exact literature search (Methods), results from the literature search (Results) and the future directions (Discussion).
Author Response
Dear Reviewer, please find our comments in blue :
This article discusses the potential applications of AI with respect to immunogenicity or non-compliance. This is an interesting article and topic, but I found it difficult to distinguish between what was background information, what related to the authors analysis of the literature and future directions. Thank you for your comment, we amended the manuscript to better explain our approach. Particularly, we provided a flowchart as requested by Reviewer 1.
The authors have provided a comprehensive discussion of their topic, but they need to provide:
1) Reasons for their selection of key words in their literature search. For example, "Nocebo effect" is introduced, but only explained later in the manuscript. Thank you for your comment, the flowchart is supported by a short description explaining the key words used resulting in inclusion of 3 sets of articles (e.g. Nocebo effect is part of the 1st set of articles regarding Biosimilar overview).
2) A flowchart of rationale for their rejection of articles (i.e. they identified 75 articles but their final number was 35). Please find the flowchart included in the method section which provide a rationale for inclusion and exclusion of articles identified.
3) A table of the articles selected, including key information such as authors, title, date, journal, country, ML algorithms used etc. Thank you for raising this point. We focused our article on Biosimilars. As a matter of fact, very few articles are related to technical aspects of artificial intelligence. We provided additional details to emphasize that our review is focused on Biosimilars and raises discussion about relevance of involving Neural Network model.
There is a great deal of repetition of information in this manuscript. For example, information about the economic incentives is repeated in the Background and in Section 5. We removed the economic incentives repetition from the Introduction section.
In Section 5 the authors refer to "Europe" but what about any other continents/countries? Europe is only mentioned as an example considering substitution rights is defined by each state.
The authors provide a sections on machine learning (7.2) and a section on neural networks (7.3), but section 7.2 is only "an example of a machine learning model". Please provide more examples of the current use with different neural network models (e.g. deep learning (DL) algorithms). Thank you for pointing this aspect out. As emphasized by Doupe et al, we decided to mention Neural Network that “Can help predict outcomes with highly complex, nonlinear relationships and interactions”. We decided to mention also that further discussion is required regarding suitability of different AI models.
This article seems to be specific to the development of a neural network model and this should be articulated in the title, abstract and aims, rather than the all-encompassing "Machine Learning" term as other algorithms are not investigated. The objective and title of the articles have been amended to reflect the focus on Biosimilar and the discussion we developed.
This is an interesting topic and the proposed model is interesting, but the manuscript would greatly benefit from harmonisation of the information, re-structure of sections to make it clearer to the reader the background (Introduction), exact literature search (Methods), results from the literature search (Results) and the future directions (Discussion). We amended the article in a way that Introduction, Methods, Results and Discussion are clearly appearing.
Please do not hesitate to reach out and share your comments.
Reviewer 3 Report
An interesting topic and an interesting application of machine learning. The article looks very superficial, non-specific.
The title does not correspond to the content, where the authors write about the review of 35 other works.
The controversy of the issue is not very systematic.
Machine learning methods are not examined in detail, as the accuracy of these methods is not addressed depending on the approaches used and specific solutions.
The conclusion does not correspond to the set goals, the conclusion is too general.
Author Response
Dear Reviewer, please find our comment in blue :
An interesting topic and an interesting application of machine learning. The article looks very superficial, non-specific. Thank you for your comment, We are aware that this article only addresses partially the involvement of Artificial intelligence regarding Biosimilar use. Our goal was to initiate a discussion about potential applications of AI in order to open further discussion about the relevance of such application. Such applications should be completed by technical reviews of AI applications that we could not cover in this article focused on Biosimilars safety.
The title does not correspond to the content, where the authors write about the review of 35 other works. You are totally right. Title has been amended accordingly. We also provided a flowchart to describe inclusion and exclusion of articles resulting in inclusion of 3 sets of articles.
The controversy of the issue is not very systematic. This article is the opportunity for us to discuss relevance of involving Artificial intelligence for Biosimilars follow up in real life. We do not have a comprehensive answer at this stage.
Machine learning methods are not examined in detail, as the accuracy of these methods is not addressed depending on the approaches used and specific solutions. Thank you for raising this point requiring discussion. As emphasized by Doupe et al, 2019, we decided to mention Neural Network that “Can help predict outcomes with highly complex, nonlinear relationships and interactions”. We decided to mention also that further discussion is required regarding suitability of different AI models.
The conclusion does not correspond to the set goals, the conclusion is too general. Thank you for your feedback, the objectives have been reviewed to focus on relevance of AI regarding Biosimilars follow up and discuss potential approaches such as Neural Network. The article is intended to shed light on the potential interest of AI in order to improve the use of biosimilars in current medical practice and should be complemented by subsequent more technical work to determine if AI can lead to an optimization of therapeutic management integrating biosimilars and biologics.
Please do not hesitate to reach out and share your comments.
Round 2
Reviewer 1 Report
The authors have attended my comments satisfactorily. The new flowchart is clarifying.
Author Response
The authors have attended my comments satisfactorily. The new flowchart is clarifying.
Thank you for your feedback. For your information, reviewer 2 provided comments regarding removing AI references. Thus, the introduction, the conclusion and the titles have been adapted to only mention Machine Learning and Neural Network.
Reviewer 2 Report
Thank you for your revisions and replies to my comments and suggestions. However, I am still concerned this manuscript does not cover enough literature and model types for artificial intelligence (AI).
There are a number of deep learning models being developed in literature, using techniques such as transfer learning, that this article does not mention. It focuses on machine learning algorithms, and in particular, neural networks. This manuscript should either remove AI references and just mention machine learning and neural networks, or expand to include the deep learning algorithms.
Author Response
Thank you for your revisions and replies to my comments and suggestions. However, I am still concerned this manuscript does not cover enough literature and model types for artificial intelligence (AI).
There are a number of deep learning models being developed in literature, using techniques such as transfer learning, that this article does not mention. It focuses on machine learning algorithms, and in particular, neural networks. This manuscript should either remove AI references and just mention machine learning and neural networks, or expand to include the deep learning algorithms.
Thank you for your comments, we understand that some amendments are required to avoid misleading content. As requested, we removed AI references. The introduction, the conclusion and the titles have been amended to only mention Machine Learning and Neural Network.
Reviewer 3 Report
This article has the potential for starting a discussion of involving Artificial intelligence for Biosimilars follow up in real life. The revision really helped to increase the scientific and practical use of this article. Thank you for all reactions from my side.
By a flowchart is clear a system of exclusion of articles resulting in the inclusion of 3 sets of articles which are discussed below.
The rework structure increased the readability of the article.
After this revision, I recommending publishing in IJERPH.
Author Response
This article has the potential for starting a discussion of involving Artificial intelligence for Biosimilars follow up in real life. The revision really helped to increase the scientific and practical use of this article. Thank you for all reactions from my side.
By a flowchart is clear a system of exclusion of articles resulting in the inclusion of 3 sets of articles which are discussed below.
The rework structure increased the readability of the article.
After this revision, I recommending publishing in IJERPH.
Thank you for your feedback. For your information, reviewer 2 provided comments regarding removing AI references. Thus, the introduction, the conclusion and the titles have been amended to only mention Machine Learning and Neural Network.